# Long Non-Coding RNAs in Cardiovascular Diseases: Potential Function as Biomarkers and Therapeutic Targets of Exercise Training

**DOI:** 10.3390/ncrna7040065

**Published:** 2021-10-11

**Authors:** Camila Caldas Martins Correia, Luis Felipe Rodrigues, Bruno Rocha de Avila Pelozin, Edilamar Menezes Oliveira, Tiago Fernandes

**Affiliations:** 1Institute of Biomedical Sciences, University of Sao Paulo, Sao Paulo 05508-030, Brazil; caldascamila13@usp.br; 2Laboratory of Biochemistry and Molecular Biology of Exercise, School of Physical Education and Sport, University of Sao Paulo, Sao Paulo 05508-030, Brazil; l.rodrigues@usp.br (L.F.R.); bruno.pelozin@usp.br (B.R.d.A.P.); edilamar@usp.br (E.M.O.)

**Keywords:** aerobic training, lncRNAs, cardiovascular disease, biomarkers

## Abstract

Despite advances in treatments and therapies, cardiovascular diseases (CVDs) remain one of the leading causes of death worldwide. The discovery that most of the human genome, although transcribed, does not encode proteins was crucial for focusing on the potential of long non-coding RNAs (lncRNAs) as essential regulators of cell function at the epigenetic, transcriptional, and post-transcriptional levels. This class of non-coding RNAs is related to the pathophysiology of the cardiovascular system. The different expression profiles of lncRNAs, in different contexts of CVDs, change a great potential in their use as a biomarker and targets of therapeutic intervention. Furthermore, regular physical exercise plays a protective role against CVDs; on the other hand, little is known about its underlying molecular mechanisms. In this review, we look at the accumulated knowledge on lncRNAs and their functions in the cardiovascular system, focusing on the cardiovascular pathology of arterial hypertension, coronary heart disease, acute myocardial infarction, and heart failure. We discuss the potential of these molecules as biomarkers for clinical use, their limitations, and how the manipulation of the expression profile of these transcripts through physical exercise can begin to be suggested as a strategy for the treatment of CVDs.

## 1. Introduction

Cardiovascular diseases (CVDs) are the leading cause of death worldwide. It is estimated that approximately 18.5 million people die annually on account of these diseases, with a third of these people dying under the age of 70 years [1]. Identifying those most affected by CVDs and ensuring they receive the appropriate treatment can prevent premature deaths. Furthermore, the development of new therapeutic strategies and biomarkers with the potential to predict the progression of CVDs is fundamental to reducing mortality worldwide. CVDs can be defined as disorders that affect the heart or blood vessels such as heart failure, coronary heart disease, cerebrovascular disease, peripheral arterial disease, and congenital heart disease [1].

The main risk factors associated with these diseases are smoking, excessive alcohol consumption, sedentary lifestyle, obesity, high blood cholesterol, among others. Individuals at risk of developing CVDs may, therefore, have increased blood pressure, glucose, and triglycerides as well as overweight and obesity. Among the many risk factors that predispose to the development and progression of CVDs, a sedentary lifestyle, supported by consistently low levels of physical activity, represents a major contributor to CVDs. On the other hand, regular exercise is associated with health benefits and a lower risk of disease [2,3]. Several studies have demonstrated that increased physical activity promotes a reduction in all-cause mortality and can increase life expectancy, affecting a strongly link to a decline in the risk of developing CVDs, in part by promoting weight loss, blood pressure control as well as improving blood lipid profile and insulin sensitivity [2,4]. For these reasons, physical activity has been recommended worldwide for CVD prevention and treatment. Despite the benefits of regular physical exercise, the molecular mechanisms by which they occur are still poorly understood.

In recent decades, a research effort has been aimed at identifying the major physiological, biochemical, and molecular contributors to the cardiovascular benefits of exercise. This research resulted in advances obtained from observational studies and interventions in both human and animal models. The Encyclopedia of DNA Elements (ENCODE) [5], a project realized in 2012, challenged the central dogma of biology and the interpretation of what is considered a functional region of the human genome [6]. From the use of high-throughput genomic platforms, it was discovered that the coding transcripts (i.e., mRNAs) represent less than 3% of the genome, while everything else represents transcripts that have little or no ability to synthesize protein. These transcripts are called non-coding RNAs (ncRNAs) [7]. For a long time, these non-coding transcripts were neglected and treated as “junk of the DNA” [8,9], “transcription noise” [10,11,12], or even as “dark matter of the genome” [13,14]; however, evidence shows that ncRNAs are not only functionally active as RNA molecules but are also one of the major regulatory networks of gene expression at the epigenetic, transcriptional, and even post-transcriptional levels (for more information, see References [9,15,16,17,18]).

According to the number of nucleotides (nt), ncRNAs can be divided into two large classes: those with fewer than 200 nt, called small non-coding RNAs such as microRNAs (miRNAs), and those with more than 200 nt, called long non-coding RNAs (lncRNAs). Among them, miRNAs are better understood and act mainly in post-transcriptional control as protein synthetic silencers binding to their target mRNA on which they induce translation degradation or repression [19]. The identification of stable miRNAs in body fluids, a strong indicator of cell–cell communication via circulating RNAs, suggested for the first time the possibility of non-coding transcripts serving as diagnostic and prognostic biomarkers for several diseases [20,21] as well as the possibility of their being used as therapeutic targets and monitoring of physical performance induced by exercise training [22,23,24,25]. On the other hand, lncRNAs can modulate gene expression at multiple levels and in an even more complex way than a regulation made by miRNAs [26,27,28]. However, lncRNAs have only recently attracted the attention of researchers, and knowledge about them, including their potential as a biomarker and therapeutic target, is still limited [29,30,31,32].

According to the LncRNA Disease v2.0 database (www.rnanut.net/lncrnadisease, accessed on 16 September 2021), there are currently more than 205,959 associations between lncRNAs and diseases including CVDs. As knowledge about these associations grows, so does the interest in investigating the influence of exercise training on the modulation of the expression of these transcripts and the possibility of using the health benefits as potential therapeutic targets [33,34,35]. In this review, after a simple presentation of lncRNAs, their biogenesis, classifications, and mechanisms of action, we discuss the current understanding of the actions performed by these ncRNAs in cardiac pathophysiology. Finally, we review the most recent publications on the role of lncRNAs in cardiac adaptations induced by exercise training, addressing the therapeutic potential of lncRNAs in clinical applications for CVDs and propose considerations regarding the present and future of research in this area.

## 2. Long Non-Coding RNAs

Differently from what was previously thought, the complexity of organisms may originate not in the number of coding genes that the species has but in the number and diversity of non-coding genes [36,37]. It is estimated that the more complex the organism, the greater the ratio between the number of ncRNAs and the number of mRNAs present in its genome [37]. Pioneers in ncRNA studies have already stated that being “junk” is not synonymous with being useless [38,39,40].

The arrival of state-of-the-art sequencing technology expanded the frontiers of knowledge by bringing with it, in the middle of the 2000s, the RNA-seq [41]. Through a large-scale sequencing methodology and bioinformatics tools, this technique allows not only to quantify RNA levels more precisely than techniques such as microarrays but also allows to trace the profile of a transcriptome, identifying which molecules of RNA are being expressed [41]. With the RNA-seq technique, it is possible to visualize the real dimension and intricacies involved behind the human transcriptome in a quick and less expensive way.

Each year, more lncRNAs are discovered, and according to version 6.0 of NONCODE, a database dedicated to ncRNAs, especially lncRNAs, the human species contains approximately 173,112 lncRNA transcripts [42], while the current number of transcripts with protein-coding capacity corresponds to approximately half, i.e., 86,757 (based on GENCODE, version 38 [43]). However, the naming, categorization, validation as well as the annotations of lncRNAs are still not complete.

## 3. Definition

The definition of lncRNAs is still ongoing. At present, there is no universal description method for these transcripts; thus, there are multiple synonyms describing similar or slightly different molecules to lncRNAs which creates some confusion as to what is or is not an lncRNA [29]. Based on the basic characteristics of lncRNAs, they are generally defined as those transcribed with more than 200 nt that do not have the capacity to encode proteins. However, such a definition that uses a criterion based on the length of the transcript has been shown to be increasingly arbitrary, as high-throughput sequencing and computational analyses contribute to the characterization of ncRNAs, since it was defined taking into account only the small ncRNA isolation protocol already well characterized [44]. Contrary to what was initially thought, some lncRNAs contain small open reading frames; that is, they have the coding potential for peptides, albeit reduced [45,46].

## 4. Biogenesis

There is a great similarity regarding the biogenesis of mRNAs and lncRNAs [28,47,48]. lncRNAs are encoded within the genome, suggesting that their transcription might be tightly coordinated with the transcription of other genes including protein-coding genes. Similar to protein-coding genes, most lncRNAs are transcribed by RNA polymerase II (RNA PolII). This enzyme, unlike RNA Pols I and III, has a high processing capacity capable of ensuring the stability of the non-coding transcript and, therefore, its functionality [29]. They also undergo post-transcriptional modifications of capping with methylguanosine at the 5’ end of their still immature transcript and polyadenylation at the 3’ end and, in various situations, they undergo alternative splicing and chemical modifications in their bases. However, since the lncRNA class houses a large diversity of transcripts, there are exceptions to those of non-canonical formations, where they do not resemble the processing of mRNAs such as circular RNAs (circRNAs) [49,50,51] and sno-lncRNAs [52].

## 5. Classification

There is still no consensus on the classification of lncRNAs [32,53], although the most common method considers their genomic location in relation to their proximity to the coding genes (Figure 1).

Then, lncRNAs can be classified as intergenic, those located between two protein-coding genes, representing the vast majority of lncRNA transcripts; intronic or exonic, they are located in the introns or exons of a protein-coding gene, respectively. These classes can also be divided into opposite forms (on the opposite strand) to a coding gene and as bidirectional promoters, those located within the promoters in the opposite direction to the encoding gene. Other studies have identified new classifications for lncRNAs such as the enhancer classification (for a review, see References [54,55]).

## 6. Mechanisms of Action

Although the primary structure is an important property for the function of a transcript, whether coding or not, the position in the genome as well as the secondary and tertiary structure of lncRNAs are better conserved interspecies than their amino acid sequence. The fact that lncRNAs exhibit poor conservation in their primary structures does not imply a loss of RNA function, rather it suggests a role in the changes in complexity a species [56]. Based on their molecular mechanisms of action, lncRNAs can be divided into signal, decoy, guide, and scaffold [28,29,57,58,59].

lncRNAs have lower levels of expression compared to mRNAs. However, lncRNAs have a more specific expression pattern than mRNAs in response to various stimuli [60,61]. Such specificity, which varies according to cell type, tissue, the subcellular compartment in which they are found, and also according to the stage of development or disease [62,63], suggests that the control of the expression of these molecules is under significant transcriptional control. Thus, lncRNAs end up playing a role as a molecular signal, signal lncRNAs [64,65,66,67,68], given that each transcript is expressed in a very specific time–space context. Once they are expressed, they can play roles as biomarkers of biological or pathological events such as the development of cardiac tissue [55,69] and CVDs [69,70], respectively. lncRNAs also could act as a molecular sponge for regulatory factors to repress transcription. To do this, the transcript simply binds to its targets, which can be RNA-binding proteins, such as transcription factors or chromatin-modifying enzymes as well as RNA sequences such as miRNAs. As a consequence, the so-called decoy lncRNAs [71,72,73,74,75,76] end up limiting the availability of regulatory factors and, therefore, preventing their interaction with their targets. Another mechanism of molecular action is based on the ability of lncRNAs to bind to regulatory proteins or to those enzymatically active, such as transcription factors and chromatin modifiers, and guide them to specific sites in the genome to control the expression of a target gene. They can also bind directly to other transcripts with regulatory functions or even to DNA itself. Thus, they guide changes in chromatin and, therefore, in gene expression in both cis and trans. The lncRNAs with this function are called guide lncRNAs [77,78,79,80,81,82,83]. They can also support the assembly of structures with several molecular components through the simultaneous binding of their domains to different transcriptional activator or repressor effector molecules. The lncRNAs that act through this mechanism are scaffold lncRNAs [84,85,86].

It is noteworthy that these mechanisms are not mutually exclusive. On the contrary, a single lncRNA is capable of acting under more than one molecular mechanism of action at the same time [57], leading to the activation or repression of gene expression at its different levels of regulation.

## 7. Functions of LncRNAs in the Heart

Studies to identify the expression of heart-specific lncRNAs revealed that some of these transcripts already present a differential expression profile in cardiovascular development as well as in pathological conditions that affect this organ [34,55,70,87,88,89,90]. The cardiogenic lncRNA cardiac mesoderm enhancer-associated non-coding RNA (CARMEN), for example, is a crucial regulator of human cardiac precursor cell fate, differentiation, and homeostasis [91]. Another important example is fetal-lethal non-coding developmental regulatory RNA (FENDRR [92]), an intergenic lncRNA with high expression in the lateral mesoderm, the germinal leaflet that gives rise to the heart. By binding to the polymeric histone remodeling complex PRC2 and TrxG/MLL, FENDRR, as an lncRNA guide, acts by modulating the chromatin state [93]. Another study involving knockout mice for a set of lncRNA, including this transcript, which has its ortholog in humans, demonstrated the importance of FENDRR not only for perfect heart development and, consequently, perfect functionality but also for embryo survival [94]. Since then, hundreds of lncRNAs have had their expression profile traced in the heart involved in a range of cellular processes, including cardiomyocytes, endothelial, and smooth muscle cells, and in fibroblasts [55,95,96,97]. Despite these findings, the regulation of cardiac pathways by lncRNAs, that is, the characterization of the functional roles of these transcripts, is still poorly understood.

## 8. LncRNAs in Cardiovascular Diseases

As mentioned before, lncRNAs can be correlated with many human diseases [98,99,100,101] including CVDs. The first association between lncRNA and heart disease came from genetic studies in which it was discovered that the locus enriched in single nucleotide polymorphisms involved with myocardial infarction susceptibility was not actually a protein-coding locus but coding for an ncRNA, which the discoverers named MIAT (myocardial infarction-associated transcript) [102]. Since then, several studies have reported associations between lncRNAs and CVDs (Table 1). 

Given the specificity of expression of lncRNAs, it would be careless to think that the dysregulation of the expression of these molecules in cardiac pathological processes, even if the molecular mechanism behind them is not exactly understood, was a mere coincidence [69]. The poor conservation of these interspecies transcripts, however, makes it difficult to translate findings in rodent models for human applications; however, several studies have shown promising results regarding the prognosis of CVDs and new therapies from the modulation of cardiac lncRNAs [33,34,70,89,90,172,173]. We summarize the lncRNAs and the CVDs (Figure 2).

## 9. Arterial Hypertension

Arterial hypertension (AH) is among the most common CVDs and is a major risk factor, if not the main one, for the development of other diseases that affect this system. Vascular smooth muscle cells (VSMCs), as a contractile element of the vessel wall, are responsible for regulating vascular tone and resistance. In this sense, abnormalities in the processes of proliferation, migration, and differentiation of these cells consequently contribute to the pathogenesis of hypertension [174]. Despite its severity, the molecular mechanism of the pathogenesis of this disease has not been precisely defined, given its complex etiology. However, although in much smaller proportions than what is currently available on miRNAs [175], few studies have identified lncRNAs specifically expressed in these cells and, therefore, involved in the pathophysiology of AH through the regulation of vascular tone [176,177].

One of these studies performed in VSMCs of rats treated with angiotensin II (Ang II)—a peptide hormone inducing vasoconstriction, inflammation, fibrosis, and hypertrophy/hyperplasia—identified a variety of lncRNAs with differential expression, among them, lnc-Ang362, one of the few lncRNAs conserved in humans. miRNAs-221 and -222, previously related to both the regulation of VSMC proliferation and the actions of Ang II on endothelial cells (ECs), were also overexpressed in response to treatment with Ang II and appeared to be co-transcribed with the lncRNA [145]. In fact, silencing lnc-Ang362 decreased the expression levels of these miRNAs as well as decreased VSMC proliferation [145]. Another experiment, this time with growth arrest-specific 5 (GAS5), an lncRNA expressed mainly in ECs and VSMCs, showed that this lncRNA, by regulating the functions of ECs and VSMCs via signaling by β-catenin, plays a fundamental role in vascular remodeling in hypertension, which is a determining process in the prognosis of the disease [124]. The GAS5 knockdown mainly aggravated the hypertensive condition in rat models spontaneously hypertensive from the increase in blood arterial pressure and, together with that, it exacerbated the pathological arterial vascular remodeling, a common sequel in hypertensive humans [124]. GAS5 has also been described as an inhibitor of PDGF-bb-induced proliferation and migration of VSMCs by competitively binding to miRNA-21 and removing the inhibitory effect of this miRNA on its target PDCD4, a crucial regulator of apoptosis and proliferation of VSMCs [125]. a growth factor and proinflammatory cytokine-induced vascular cell-expressed RNA (Giver) is another example of an lncRNA that by induction of Ang II via the NR4A3 nuclear receptor mediates the expression of inflammatory genes and promotes increased oxidative stress and proliferation of VSMCs [129]. In the arteries of hypertensive patients related to this study, Giver had a significantly elevated expression profile, whereas when these patients were treated with angiotensin-converting enzyme inhibitors and angiotensin receptor blockers, a marked reduction in this lncRNA was observed [129].

Another study identified AK098656 as an lncRNA responsible for promoting increased proliferation and migration of VSMCs, that is, the synthetic phenotype of VSMCs, through increased synthesis of matrix proteins and degradation of contractile proteins in hypertensive patients [105]. According to the study, this lncRNA mediates the stability of cytoskeletal proteins through its scaffold mechanism to induce VSMCs differentiation and, consequently, causes resistance artery remodeling thus leading to an increase in blood pressure. In vivo, mice overexpressing this lncRNA spontaneously developed hypertension, marked by the synthetic VSMCs phenotype and the narrowing of resistance arteries [105]. These same rats also appeared to be suffering from mild cardiac hypertrophy [105]. The results of a genetic study in Turkish hypertensive patients suggested that polymorphisms rs10757274, rs2383207, rs10757278, and rs1333049 within the lncRNA antisense non-coding RNA in the INK4 locus (ANRIL), also known as CDKN2B-AS1, could confer increased susceptibility to the development of AH [107].

## 10. Coronary Heart Disease

Atherosclerosis is a chronic, progressive inflammatory process common in several CVDs and is characterized by the formation and deposition of fibrofatty plaques in the wall of arteries [178]. This accumulation of atheromatous plaques results in the narrowing and stiffening of these vessels, such that, over time, blood flow is partially or totally blocked, leading to tissue ischemia [178]. When atherosclerosis occurs in the arteries that supply the heart (i.e., the coronary arteries), it is then called coronary heart disease (CHD). Considering that several lncRNAs are responsible for conducting the functions of ECs, VSMCs, vascular inflammation, and cell metabolism, the dysregulation of the expression of specific non-coding long transcripts is involved with the development and progression of atherosclerosis and, consequently, CHD [179].

Among the findings, it was reported that high levels of H19 expression were associated with the diagnosis of CHD [133] and linked to hyperhomocysteinemia, an important risk factor for the development of CHD [180], and to polymorphisms also associated with the risk of developing CHD in the Chinese population [134]. The ANRIL lncRNA, discovered from genomic association studies, is yet another example of a transcript identified as directly related to the risk and severity of this disease. It is suggested that SNPs in this lncRNA contribute to susceptibility to CHD [106]. In VSMCs, lincRNA-p21 undoes the link between p53, which has a transitional target, and its mouse double minute 2 inhibitor (MDM2), leading to regulatory effects on cell proliferation and apoptosis [83]. In patients with CHD, this lncRNA has been shown to have low expression in coronary artery tissues [181]. Furthermore, polymorphisms in lincRNA-p21 were also associated with an increased risk of CHD [142]. A Brazilian study suggested the involvement of MIAT and MALAT1 in the pathogenesis of CHD in the Brazilian population [149]. According to the data obtained, both lncRNAs demonstrated high plasma expression levels in patients with CHD [149]. However, while MIAT, which has previously been associated with vascular endothelial dysfunction, does appear to be related to CHD, the data on MALAT1 is still inconclusive.

More examples of lncRNAs that have been shown to be associated with CHD are BRAF-activated non-protein coding RNA (BANCR) [115], cholesterol homeostasis regulator of miRNA expression (CHROME) [121], HOXA transcript at the distal tip (HOTTIP) [138], LINC00968 [143], nexilin F-actin binding protein antisense RNA 1 (NEXN-AS1) [160], and smooth muscle-induced lncRNA enhances replication (SMILR)m [165].

## 11. Acute Myocardial Infarction

Acute myocardial infarction (MI) is an interruption in blood flow, in any coronary artery, that induces a reduction in blood supply. These interruptions induce some portion of the cardiac muscle tissue to suffer an ischemic process, receiving less oxygen to meet their needs leading to metabolic diseases. It has been demonstrated that there are lncRNAs directly or indirectly involved in the occurrence of MI and that their expression levels have potential signaling for the disease [108,158].

MIAT has been characterized as an lncRNA with a pro-fibrotic function after MI [156]. Acting like a sponge on important miRNAs with anti-fibrotic function, such as miRNA-24, the action of MIAT not only led to myocardial degeneration by fibrosis but also, consequently, promoted adverse remodeling in the heart of infarcted rats [156]. On the other hand, the MIAT knockdown prior to MI contributed to the preservation of cardiac function by decreasing the extent of matrix deposition and interstitial fibrosis [156]. Other publications have now raised the pro-hypertrophic potential of MIAT in cardiomyocytes from its action as a sponge on miRNAs-150 [157] and -93 [182]. In a mouse model, shortly after MI induction, it was identified that two lncRNAs had a significantly high level of expression in the heart: myocardial infarction-associated transcript 1 (Mirt1) and 2 (Mirt2) [158]. Unlike MIAT, these transcripts have no direct relationship with infarct size and seem to play a protective role in cardiac ventricular function through intracellular communication via paracrine signaling. Its expression levels are correlated with the expression of genes known to be involved with the preservation of the ejection fraction and, therefore, with processes that affect left ventricular remodeling after MI such as extracellular matrix turnover, fibrosis, apoptosis, and mainly inflammation [158,183].

Some others lncRNAs have been proposed as MI biomarkers. The dysregulation of plasma LIPCAR levels in samples from patients with ventricular remodeling after infarction, for example, suggests an association, although it may not be causal of this lncRNA with the disease [140]. The finding that lncRNA urothelial cancer-associated 1 (UCA1) stimulates cardiomyocyte apoptosis, contributing to the progression of cardiac injury in rats with cardiac injury induced by ischemia-reperfusion injury, raised interest in its potential as a biomarker and also as a potential therapeutic target [167,168]. In the cardiac response to the ischemic process, the lncRNA aHIF appears as an mRNA destabilizer, and its translation product is the hypoxia-inducing factor 1α (HIF-1α), the master regulator of the cellular response such as angiogenesis and hypoxia [103,108]. In the blood of patients affected by MI, aHIF was highly expressed in relation to its expression levels in healthy patients [108]. The finding, consistent with the fact that HIF-1α is induced by hypoxia, suggests that there is likely an interaction between the mRNA HIF-1α and the lncRNA aHIF in regulating the post-ischemic angiogenesis process [103,108]. In MI, the inhibition of the autophagic process in an ischemic context can play a cardioprotective role itself, given that, from the moment that autophagosome accumulation is observed, especially when in reperfusion, autophagia ceases to be cardioprotective and starts to promote the death of cardiomyocytes. With low expression in the plasma of these patients, HOX transcript antisense RNA (HOTAIR) is another example of an lncRNA with a cardioprotective function in MI and a potential predictor for the diagnosis of MI [137]. HOTAIR regulates apoptosis through the sponge mechanism over miRNA-1, an miRNA already known to be at low levels of expression in infarcted myocardial tissue [137].

During ischemia-reperfusion injury, autophagy promoting factor (APF), highly expressed in this scenario, was responsible for the sequestration of miRNA-188-3p [112]. This miRNA targets the ATG7 gene, an autophagy promoter. miRNA sequestration, therefore, promoted the upregulation of ATG7 expression [112]. This study proposed APF and miRNA-188-3p as therapeutic targets for autophagy inhibition aiming at cardioprotection in MI [112]. Moreover, in this context of cell death, several studies have shown the participation of lncRNAs in regulating the death of viable cardiomyocytes by apoptosis in MI. Cardiac apoptosis-related lncRNA (CARL) [116] and mitochondrial dynamic-related lncRNA (MDRL) [150] are examples of these lncRNAs and were transcriptionally repressed after MI. The overexpression of these in vivo resulted in smaller-sized infarcts. The same consequence was observed from the silencing of Wisp2 super-enhancer-associated RNA (Wisper) after MI [170]. CARL and MDRL, by inhibiting miRNA-539 or miRNA-361, which are pro-apoptotic miRNAs, respectively, inhibited mitochondrial fission and apoptosis of cardiomyocytes. Like these lncRNAs, 5 prime to Xist (FTX), after reperfusion injury and ischemia, was found to have low expression levels and, in vitro, was described with anti-apoptotic action through the regulation of the apoptosis repressor BCL2L2 via sequestration of miRNA-29b-1-5p [123]. 

Many other lncRNAs have been implicated in the pathophysiology of MI, including ANRIL [108,109,110], CDR1 Antisense RNA (CDR1AS) [117], GAS5 [126,127,128], H19 [130,131,132], KCNQ1OT1 [108], lnc-Ang362 [146], metastasis associated lung adenocarcinoma transcript 1 (MALAT1) [108,132,147], maternally expressed gene 3 (MEG3) [151], myosin heavy chain-associated RNA transcripts (MHRT) [153], n379519 [159], necrosis-related factor (NRF) [163], NONRATT021972 [161], pro-cardiac fibrotic lncRNA (PCFL) [164], testis-specific transcript Y-linked 15 (TTTY15) [166], upregulated in hypothermia-treated cardiomyocytes (UIHTCs) [169], and zinc finger antisense 1 (ZFAS1) [117,171,184]. Therefore, the intrinsic relationship between lncRNAs and MI is evident.

## 12. Heart Failure

Heart failure (HF) refers to the heart’s inability to adequately perform its function as a blood pump due to the fact of several pathological processes of cardiac remodeling that affect its ability to contract and/or relax [185,186]. Similar to what has been reported for the other CVDs mentioned above, several lncRNAs have been identified behind the mechanisms involved with the progression of maladaptive cardiac remodeling in HF [111,172,187].

Myosin heavy-chain-associated RNA transcripts (MHRT) is a cluster of specific lncRNAs that partially overlap the Myh7 locus, which by harboring the human gene encoding β-myosin heavy chain is critical for cardiac function as an ejector pump [154,188]. MHRT is highly expressed in the heart under physiological conditions, given its protective role in reducing or even inhibiting the expression of genes involved with changes in pathological aspects of heart muscle contractility [154,155]. However, under stress conditions, such as what occurs in HF induced by transverse aortic constriction, a significant decrease in MHRT expression was reported and, concomitantly with a change in the Myh6 isoform (α) for Myh7(β), a well-known feature of HF development [154,189]. Under physiological conditions, MHRT inhibits its silencing by directly interacting with Brg1, the chromatin remodeling factor, in the region of its bidirectional promoter shared with Myh6 [154]. However, under cardiac stress situations, the expression of Brg1 exceeds that of MHRT, so that Brg1 is free to promote its remodeling function on chromatin and, thus, promote the exchange of the heavy chain from α-myosin to β-myosin [154]. The role of MHRT as a protector is validated when, from the restoration of its expression level, the protection of the heart against pathological hypertrophy and, therefore, against HF is noticed [154].

With a specific function in cardiomyocytes, cardiac hypertrophy-related factor (CHRF) was the first lncRNA to be related to HF. In in vitro experiments in cardiomyocytes, overexpression of CHRF induced a pathological process of hypertrophy and, in vivo, it induced apoptosis [120,139]. By sequestering the miRNA-489, this lncRNA prevents the interaction of the microRNA with its target Myd88 (myeloid differentiation primary response gene Myd88), an indispensable gene in the regulation of cardiac hypertrophy [120]. The hypertrophy-associated epigenetic regulator (CHAER) cardiac lncRNA, abundantly expressed in the heart of mice with HF, is another transcript involved in the control of cardiac remodeling [118]. This lncRNA interacts with PCR2 with the purpose of inhibiting the methylation of lysine residues in histones that are found in the promoter regions of pro-hypertrophic genes, making them amenable to expression. When silenced, it was observed that the pathological cardiac hypertrophic process was alleviated. Similarly, cardiac hypertrophy-associated transcript (CHAST) is characterized as a pro-hypertrophic lncRNA [119]. In both mouse hypertrophic cardiac models and in the cardiac tissue of patients with aortic stenosis, characterized by the compensatory development of cardiac hypertrophy, it was observed that CHAST was highly expressed [119]. In these mice, CHAST overexpression induced cardiomyocyte hypertrophy both in vivo and in vitro, while its suppression proved cardio-protective by preventing hypertrophy and preserving cardiac function in these animals [119].

Some of the signaling mechanisms mediated by lncRNAs participating in different pathological processes in the heart and vascular tissues in heart failure, myocardial infarction, coronary heart disease, and arterial hypertension are depicted in Figure 3.

## 13. LncRNAs as Biomarkers and Potential Therapeutic Targets for Cardiovascular Diseases

Offering a treatment that specifically meets the needs of a patient is the goal of personalized cardiovascular medicine, which considers aspects, such as clinical, genetic, genomic, and even environmental data, related to the patient in question [190,191,192]. However, despite technological advances, the current tools and methods available still lack precision. Circulating biomarkers have been shown to be considerable indicators regarding disease progression, as they simplify and guide clinical decisions so that medical intervention is no longer a generalized pattern applied to any patient and begins to adapt to the individual’s needs. Traditionally, proteins and peptides derived from the heart are the main circulating molecules to be explored in terms of their usefulness as biomarkers for the diagnosis of CVDs, so much so that, routinely, the expression of troponin T and B-type natriuretic peptide are investigated for diagnosis or prognosis [193]. Currently, a range of studies have proposed the use of circulating mRNAs and miRNAs as cardiac biomarkers in personalized medicine [20,21,194,195,196]. Recently, since lncRNAs are differentially expressed in a specific time–space context, participate in intracellular communication, and are associated with CVDs, it has been suggested that these lncRNA may be the newest class of non-coding biomarkers for non-invasive clinical use for the diagnosis and prognosis of CVDs. In support of this possibility, it has already been shown that lncRNAs can be detected in extracellular body fluids such as serum, plasma, and urine [102,140,197,198,199,200]. In addition to being encapsulated in exosomes, extracellular vesicles [122,201], and apoptotic bodies [202], circulating lncRNAs can associate with proteins and maintain the stability of their conserved transcript. Therefore, they have decisive characteristics of a biomarker: easy access, stability, and specific expression patterns in the context of CVDs.

The first study to test the use of an lncRNA as a biomarker for CVDs proposed LIPCAR as an example, since its plasma levels were shown to be related to left ventricular remodeling soon after MI as well as to the increased chances of progression to a condition of HF [140]. Those patients who developed HF as a result of an MI had a high level of LIPCAR expression in plasma compared to those patients who after the MI did not suffer from ventricular remodeling [140]. Furthermore, the magnitude of LIPCAR expression was also shown to be associated with the prediction of cardiovascular mortality in those patients with chronic HF [140]. More recently, the non-coding repressor of NFAT (NRON) and MHRT were proposed as independent predictive factors for HF, as they were found in high plasma levels in patients affected by this pathology [155]. Furthermore, others lncRNAs, such as ANRIL [111], antisense transcript of β-secretase-1 (BACE1-AS) [114], eosinophil granule ontogeny transcript (EGOT) [111], H19 [135], heart disease-associated transcript 2 (HEAT2) [136], HOTAIR [111], long intergenic non-protein coding RNA, regulator of reprogramming (lincRNA-ROR) [144], LOC285 [111], ribonuclease mitochondrial RNA processing (RMRP) [111], Ro60-associated Y5 (RNY5) [111], SOX2 overlapping transcript (SOX2-OT) [111], and steroid receptor RNA activator 1 (SRA1) [111] showed variations in their expression levels in the context of HF, which eventually suggests their potential as biomarkers for this disease.

It is not only in HF that lncRNAs have been found with the potential to serve as a biomarker. Even though the biomarker potential of lncRNAs in arterial hypertension is still a little explored subject [176], some circulating lncRNAs, such as GAS5, AK098656, NR_027032, NR_034083, and NR_104181, were differentially expressed in hypertensive patients compared to healthy individuals, suggesting a potential role of these lncRNAs as biomarkers in this disease. GAS5 is an lncRNA expressed mainly in ECs and VSMCs and was found with low expression in the plasma of hypertensive patients and in arteries and retinas of a spontaneously hypertensive rat model [124]. On the other hand, the lncRNA AK098656 stood out as positively regulated in the plasma of hypertensive patients and, mainly, the predominance of its expression was located in VSMCs [105]. Compared with normotensive patients, the lncRNAs NR_027032 and NR_034083 presented with significantly low levels in peripheral blood leukocytes from patients with essential hypertension, whereas the opposite was observed for the levels of NR_104181; that is, this lncRNA presented itself with significantly reduced levels in these patients [162]. In CHD, CoroMarker was also proposed as a promising biomarker, since it was found to be dysregulated, but stabilized in vesicles, in plasma samples from these patients [122]. aHIF [104], APOA1 antisense RNA (APOA1-AS) [104], lncRNA associated with poor prognosis of hepatocellular carcinoma (AWPPH) [113], KCNQ1 opposite strand/antisense transcript 1 (KCNQ1OT1) [104], and LIPCAR [141] are others prominent lncRNAs to be used in clinical settings as biomarkers for diagnosis and monitoring of CHD. The molecular mechanism of these lncRNAs as biomarkers in the pathophysiology of CHD, however, is still something to be better explored. In another study, this time involving peripheral blood mononuclear cells from patients with MI, MIAT presented with reduced levels of expression and was associated with ST-segment elevation in these patients, suggesting the potential biomarker of this lncRNA [197]. In addition, lncRNAs can be detected in whole blood samples from patients with MI [108]. The study responsible for this finding identified that ANRIL and KCNQ1OT1 could be used in favor of disease prognosis as predictors of ventricular dysfunction after MI [108]. Despite the evident incremental advantage to be conferred by lncRNAs in addition to the already available biomarkers, there is still much more work to be conducted until they reach clinical application.

Considering all of the above regarding the regulatory role of lncRNAs in the cardiovascular system and the observation that the dysregulation of the expression of these transcripts has been observed in CVDs, it seems appropriate to consider them as molecular targets in therapies [173,203,204,205]. Through counter-regulation, the restoration of physiological expression levels of lncRNAs can occur both through gain-of-function strategies (for those lncRNAs that are significantly impaired) and through loss-of-function strategies (for those overexpressed). Among the strategies involved in decreasing RNA expression levels is the RNA interference technique, which, using small interfering RNAs (siRNAs), is able to decrease the expression levels of lncRNAs located mainly in the cytoplasm [206,207] and, alternatively, with the use of siRNAs, there are GapmeRs that can be used in a technique based on the complementarity of single-stranded antisense oligonucleotides in the gene to be silenced [33,152,170,208,209,210]. In the opposite direction, plasmids and viral vectors have been used for the overexpression of lncRNAs [116,150,211]. Recently, gene-editing tools, such as the CRISPR system, have been proposed as attractive approaches in terms of their high therapeutic potential [212,213]. In fact, the modulation of expression levels of cardiac-specific lncRNAs in vivo has shown promising results in terms of improving cardiac dysfunction and even delaying the progression of other pathological processes in the heart already affected by some diseases. Despite this, there are obstacles to be overcome in the process of translating these lncRNA modulation strategies for the treatment of CVDs addressed here: (i) drug delivery mechanisms, which vary according to the technique used, to the heart, vessel, specific cell type, or even cell compartment; (ii) affinity of the therapeutic molecule to the target lncRNA as well as the integrity of its stability in circulation; (iii) pharmacokinetic and pharmacodynamic characterization of the drug in question; (iv) assessment of the systemic consequences of target modulation; (v) evaluation of the effectiveness and duration of responses. Resolving these issues is critical to the success of lncRNAs as new therapeutic targets for the treatment of CVDs [214].

## 14. Cardiac LncRNAs and Exercise Training 

Regular physical exercise is associated with numerous protective and restorative effects on cardiovascular health [215,216,217,218,219,220], in addition to improving the prognosis of patients after cardiovascular events. Physical exercises encompass modifiable variables [218]. These include a modality (e.g., aerobic or resistance training, frequency, intensity, and duration of exercise activities), each of which affect metabolic and molecular responses. Aerobic (or resistance-based) and resisted (or strength-based) exercises represent two extremes of the diversity of existing modalities. Aerobic exercise imposes a high-frequency (repetition) and low-power (load) demand, whereas resistance exercise imposes a low-frequency and high-load demand. Therefore, the metabolic and molecular responses to different modalities are distinct and, therefore, the specificity of a given molecular response is coupled with a functional result. Aerobic physical training has been the most investigated in the literature and with more effective results than resistance training for the treatment of the main cardiovascular risk factors, as it promotes blood pressure control, improves blood lipid profile and insulin sensitivity as well as improvements in an individual’s capacity for maximum oxygen consumption [218]. More recent studies, however, have shown that combined aerobic exercise and resistance exercise training appears to be more effective in modulating CVD risk factors than just those based on a single modality [221,222], in addition to promoting compensatory physiological adaptations capable of altering the structure of the myocardium, thus improving cardiac functionality [223]. The beneficial effects of exercise to the cardiovascular system have been well described in many excellent reviews [224,225,226,227]. 

Aerobic exercise has been characterized as the main therapeutic alternative for hypertensive patients in view of its beneficial effects on all systems involved in blood pressure regulation [228,229]. In the cardiovascular system, for example, aerobic exercise promotes an increase in cardiac output by inducing left ventricular hypertrophy of the physiological type and a reduction in heart rate that leads to a reduction in the pressure of the coronary flow on the organ, not to mention its influence on risk factors related to the development and progression of this disease [228]. In this way, aerobic exercise promotes significant improvements in the cardiac health of hypertensive patients, delaying or even preventing adversities arising therefrom. It is not only in hypertension that exercise is successfully applied for therapeutic purposes. In order to assess the effect of exercise-based cardiac rehabilitation, a large meta-analysis involving more than 8000 patients with CHD undergoing different randomized clinical training studies observed a significant 26% decrease in cardiac mortality in these patients [230,231]. If in CHD, an important precursor of MI, if not the main one, exercise already has a significant impact on cardiac mortality, it could not be different in the context of MI. After MI, physical exercise is a great ally in preventing complications resulting from infarction and in increasing the quality of life and longevity of these individuals [232,233,234,235]. Furthermore, a variety of physical training studies in patients with HF have proven the efficacy and safety of aerobic training in these patients [236,237,238,239]. Furthermore, meta-analysis studies confirmed improvements in clinically relevant parameters for HF [240,241,242]. Although there is no doubt about the restorative role of exercise on the cardiovascular system, little is known about the molecular mechanisms behind these effects.

Understanding the molecular pathways involved behind exercise-induced cardiac adaptation provides potential therapeutic targets for the treatment of CVDs [2]. Widely recommended as the most effective non-pharmacological strategy for the prevention and treatment of CVDs, exercise training therefore represents another method to be applied, aiming at modulating the expression of lncRNAs [28,35]. Thus far, only two articles have investigated the role of exercise training on the modulation of lncRNAs in the context of CVDs [148,243] (Figure 4). 

In one of these studies, it was suggested that aerobic exercise on a treadmill promoted, by decreasing MALAT1 expression levels [244], not only improvements in cardiac function and left ventricular remodeling in rats with chronic HF but also led to a reduction in the levels of reactive oxygen species and inflammatory cytokines, decreased apoptosis of cardiomyocytes, and increased autophagic proteins [148]. The overexpression of MALAT1 via plasmid vector nullified all the positive effects promoted by aerobic exercise in rats with chronic HF [148]. As a molecular mechanism, it was observed that, in vitro, MALAT1 overexpression was related to decreased miRNA-150-5p expression levels, resulting in increased cardiomyocyte apoptosis and decreased autophagy as well as suppression of the signaling pathway PI3k/Akt [148].

Aerobic exercise, also performed on the treadmill, promoted improvements in contractility and cardiac function as a pump through the improvement in left ventricular function and led to a reduction in apoptosis of cardiomyocytes and fibrotic areas in the heart of rats after MI through the normalization of levels of expression of H19, MIAT, and GAS5. H19 is an lncRNA associated with an increase in the area of infarction and fibrosis in the stages immediately after MI [130,131]. With significant expression in cardiac fibroblasts, H19 exerts its mechanism through binding to the Y-box binding protein 1 thus preventing collagen production [131]. In this sense, the positive effect of exercise on reducing fibrosis and improving cardiac function can be attributed, albeit partially, to the modulation of H19 expression levels. MIAT, in turn, is an lncRNA with pro-fibrotic action in the pathogenesis of MI that acts by sequestering miRNA-24 and, in this way, removes the inhibition of Furin, the signaling activator of the TGF-beta pathway [156]. Therefore, it is to be expected that their expression levels are high in this context. By decreasing MIAT expression levels, aerobic exercise attenuated the effects promoted by the dysregulation of this lncRNA in cardiac fibrosis. In a previous study, induction of lncRNA GAS5 expression was related to inhibition of sema3a [128]. Moreover, corroborating this finding, physical exercise promoted a significant increase in GAS and, thus, promoted beneficial effects in the rat model used in the study. Despite these findings, the researchers concluded that further investigation into the targets of these lncRNAs is needed to trace the exact molecular mechanism that links exercise to lncRNAs in MI. 

## 15. LncRNAs in Cardiovascular Diseases: Challenges and Future Perspectives

lncRNAs have characteristics of great interest to the biomedical community. These characteristics have received attention, albeit timidly, in recent clinical trials (NCT04189029; NCT03268135; NCT02915315; NCT03279770) with the purpose of investigating the role of these transcripts as biomarkers and in the pathogenesis of some CVDs. However, one cannot fail to mention the challenges to be overcome until these transcripts finally move from research to clinical application.

The isolation, detectability, quantification, and strategy adopted for the normalization of circulating lncRNAs are key factors for the reliable identification of candidates as potential biomarkers and, in the absence of standardization, have been technical limitations of important relevance for ncRNAs in general [202,214,245]. Added to this is the fact that there may be significant variations in the expression levels of lncRNAs, including those that are significant regarding CVDs, among different body fluids such as serum, plasma, and urine or even among different compartments of the same cell [199,200]. The lack of standardization regarding the fluid to be considered as a sample for a given lncRNA may end up leading to research with wrong conclusions. In addition, cardiovascular risk factors, medication use, sex, and age are examples of some factors capable of promoting changes in the expression levels of ncRNAs such as lncRNAs [196,202]. Among the limitations found in the process until lncRNAs reach the clinic stage as therapeutic targets is the fact that lncRNAs are still in the process of characterization and annotation; even at these stages, many challenges need to be overcome. In this sense, the modulation of an lncRNA can result in opposite effects, even harmful, for the purpose in question [34], since the same lncRNA can be involved in the mechanism of different pathologies [246,247]. The availability of information on the characterization and annotation of lncRNAs, therefore, provides greater knowledge about the lncRNA in question and, consequently, of other molecules with which it may be related, which allows a broader notion about the implications involved in the modulation of one of these transcripts. Furthermore, it is well known that the intermediate step between basic research and clinical trials necessarily involves the use of animal models. At this point, the lack of conservation of the nucleotide sequence of lncRNAs among different species represents a limitation with great impact, as it makes it difficult to transpose the results obtained in preclinical studies to humans [248]. Therefore, clinical trials end up being restricted to working only with those lncRNAs that have their counterparts in humans. In addition to these challenges, there is still a need to elucidate the secondary and tertiary structures of lncRNAs, which are even more critical for the function of these transcripts than the primary structure, and these may have structural homologues in other species including those used as models of experimental animals [249,250]. Finally, there are still challenges regarding drug delivery to the target lncRNA of interest [251].

Considering the limitations mentioned here, an initiative by the scientific community is needed to reach a consensus on the methods (from the way the sample is manipulated to the chosen normalization strategy) to be used in order to ensure robust paths for identifying lncRNAs as CVD biomarkers as well as precision regarding the criteria and parameters to be adopted for the formation of groups involved in future clinical studies, whether for the identification of lncRNAs as biomarkers or for the assessment of their potential as a therapeutic target. In this regard, there is still much to be overcome regarding the challenges of the application of lncRNAs as a therapeutic approach in CVDs; the clinical trials presented here refer mostly to the use of these molecules as biomarkers, some of which also investigated the role of lncRNAs in CVDs. Although important experiments using highly sophisticated technological tools, such as RNA-seq, have been conducted to identify therapeutic candidate lncRNAs, little has been done regarding the characterization of these transcripts found in terms of regulation of the pathological process or ability to undergo regulation. It is necessary, therefore, that new information that arises about a lncRNA already known or recently discovered can be accessed by any researcher, anywhere in the world, as this ensures optimization in the field of research on lncRNAs. Even today, the lack of gene homology is an obstacle in science. Future technological advances are expected to provide solutions to overcome these and other limitations that challenge the use of lncRNAs as therapeutic targets in CVDs [34,202,214,251]. Once overcome, the benefits for patients affected by CVDs can be enormous.

## 16. Conclusions

This new class of non-coding transcripts playing regulatory roles in various diseases is the beginning of knowledge. Although the number of lncRNAs discovered over the years has increased, so far very little is known about the mechanisms of action and functions performed by these molecules. One of the reasons for this delay is the poor sequence conservation of interspecies lncRNAs, as variations in different animal models make the identification of biological functions and mechanisms of action of the vast majority of lncRNAs and the consequent translation of findings from animals to humans difficult. In this aspect, databases (for example, LNCipedia [252], LncTar [253], and LncRNAWiki [254,255]) have been important tools for depositing and rationalizing information about lncRNAs from different parts of the world. Despite the challenges, lncRNAs are promising candidates for therapeutic use and are characterized as a tool with great application power in personalized medicine given their specific expression pattern associated with different pathologies. It is still the beginning of this new field of study involving the modulation of the expression of lncRNAs in the context of CVDs and physical exercise. There are, therefore, great expectations regarding the application of alternative modalities to aerobic exercise to modulate the lncRNAs involved in this context, such as resistance training and also combined training. Before therapeutic application, further research is needed for a complete functional characterization of lncRNAs involved in cardiovascular pathology as well as their ability to be regulated from different physical training protocols.

## Figures and Tables

**Figure 1 ncrna-07-00065-f001:**
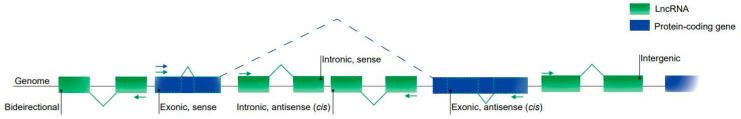
Classification of lncRNAs based on their genomic location. Intergenic lncRNAs are located between two protein-coding genes. Exonic and intronic lncRNAs are located entirely in the exonic and intronic regions of a protein-coding gene, respectively. Sense lncRNAs are transcribed from the same strand and in the same direction as the protein-coding genes; they can be both exonic and intronic. Antisense lncRNAs are transcribed from the opposite strand of the protein-coding genes and can also be both exonic and intronic. Bidirectional lncRNAs are located within the promoter region of a protein-coding gene.

**Figure 2 ncrna-07-00065-f002:**
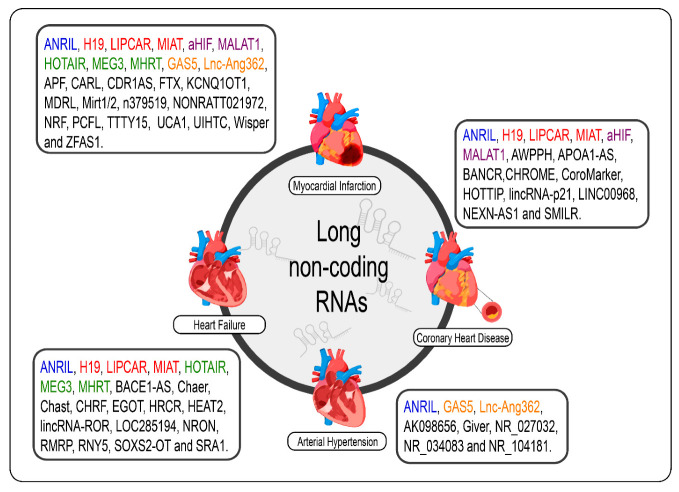
lncRNAs are differentially expressed in cardiovascular diseases such as heart failure, myocardial infarction, coronary artery disease, and arterial hypertension. The lncRNAs marked in blue are the same present in myocardial infarction, coronary artery disease, heart failure, and arterial hypertension; those marked in red are the same present in myocardial infarction, coronary artery disease, and heart failure; those marked in green are the same present in myocardial infarction and heart failure, those marked in orange are present in myocardial infarction and arterial hypertension; and the purple present in myocardial infarction and coronary artery disease.

**Figure 3 ncrna-07-00065-f003:**
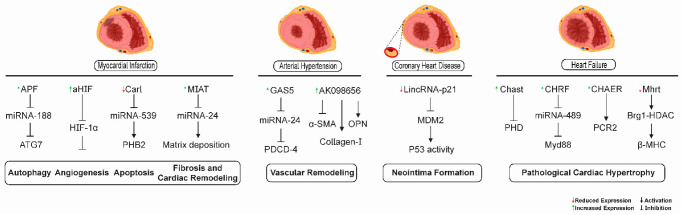
lncRNA-mediated signaling mechanisms as possible therapeutic targets for cardiovascular diseases. lncRNAs have been shown to participate in different pathological processes in the heart and vascular tissues in heart failure, myocardial infarction, coronary heart disease, and arterial hypertension. lncRNAs APF, CARL, aHIF, and MIAT were involved in autophagy, angiogenesis, apoptosis, and cardiac fibrosis processes related to myocardial infarction, respectively. The lncRNAs GAS5 and AK0986656 promoted vascular remodeling in arterial hypertension. The lncRNA lincRNA-p21 was related to neointima formation in coronary heart disease. The lncRNAs MHRT, CHAER, CHRF, and CHAST could serve as promising targets for the retardation of pathological cardiac hypertrophy in heart failure.

**Figure 4 ncrna-07-00065-f004:**
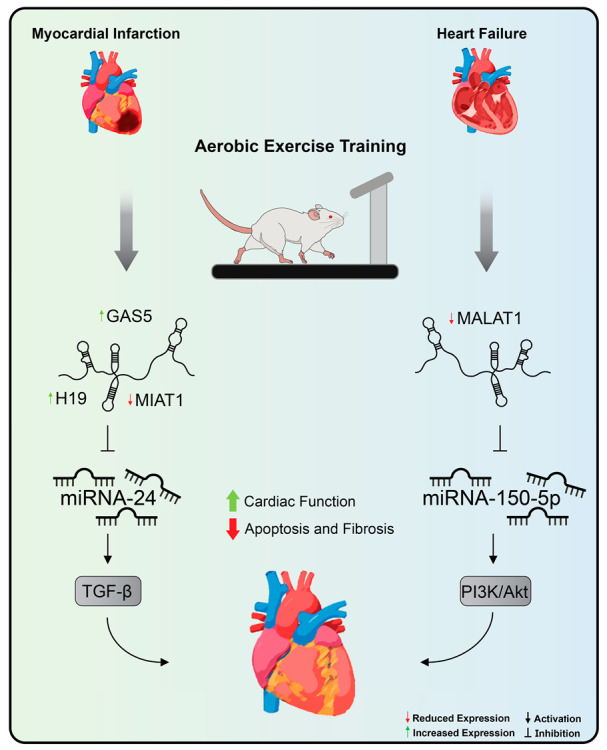
Aerobic exercise training’s effects on lncRNA expression in cardiovascular diseases. Treadmill exercise promoted, by decreasing MALAT1 and increasing miRNA-150 expression levels, improvements in cardiac function and left ventricular remodeling in rats with heart failure. In addition, aerobic exercise promoted a reduction in apoptosis of cardiomyocytes and fibrotic areas in the heart of rats after myocardial infarction through the normalization of levels of expression of H19, MIAT, and GAS5. Exercise training inhibits MIAT expression, a lncRNA with pro-fibrotic action in the myocardial infarction that acts by sequestering miRNA-24 and stimulating the TGF-beta pathway. MALAT1: metastasis-associated lung adenocarcinoma transcript 1, MIAT: myocardial infarction-association transcript, and GAS5: growth arrest specific 5.

**Table 1 ncrna-07-00065-t001:** List of lncRNAs involved in cardiovascular diseases.

lncRNA	CVDs	Association	References
aHIF	MI	Regulation of the angiogenesis process and a biomarker.	[103]
aHIF	CHD	Biomarker.	[104]
AK098656	AH	Regulation of arteries of resistance and a biomarker.	[105]
ANRIL	CHD	Susceptibility conferred by SNPs in the ANRIL locus on chromosome 9p	[106]
ANRIL	AH	Increase of susceptibility to higher systolic blood pressure conferred by polymorphisms.	[107]
ANRIL	MI	Protection of cardiomyocytes from hypoxia by acting on the miRNA-7-5p/SIRT1 axis; and biomarker to LV dysfunction.	[108,109,110]
ANRIL	HF	Biomarker.	[111]
APF	MI	Promotion of cardiomyocytes autophagy acting as a sponge for miRNA-188-3p.	[112]
APOA1-AS	CHD	Biomarker.	[104]
AWPPH	CHD	Biomarker.	[113]
BACE1-AS	HF	Promotion of ECs apoptosis.	[114]
BANCR	CHD	Promotion of VSMCs proliferation and migration.	[115]
CARL	MI	Reduction of mitochondrial fission and apoptosis acting as a sponge for miRNA-539.	[116]
CDR1AS	MI	Biomarker.	[117]
Chaer	HF	Induction of Pathological cardiac remodeling.	[118]
Chast	HF	Induction of Pathological cardiac remodeling.	[119]
CHRF	HF	Endogenous sponge to miRNA-489 activity.	[120]
CHROME	CHD	Regulation of cellular cholesterol homeostasis.	[121]
CoroMarker	CHD	Biomarker.	[122]
EGOT	HF	Biomarker.	[111]
FTX	MI	Regulation of cardiomyocytes apoptosis acting as a sponge for miRNA-29b-1-5.	[123]
GAS5	AH	Regulation of ECs and VSMCs function acting as endogenous RNA competing of miRNA-21; and a biomarker.	[124,125]
GAS5	MI	Protection of cardiomyocytes against hypoxic injury acting as a sponge for miRNA-142; promotion of the development and progression of the disease acting on the miRNA-525/CALM2 axis; and improves apoptosis by negatively regulating sema3a.	[126,127,128]
Giver	AH	Promotion of VSMCs dysfunction.	[129]
H19	MI	Induction of cardiac remodeling; autophagy; and biomarker.	[130,131,132]
H19	CHD	Biomarker.	[133,134]
H19	HF	Regulation of cardiac hypertrophy; and a biomarker.	[111,135]
HEAT2	HF	Biomarker.	[136]
HOTAIR	MI	Induction of cardioprotective acting as a sponge for miRNA-1 and as a biomarker.	[137]
HOTAIR	HF	Biomarker.	[111]
HOTTIP	CHD	Promotes ECs proliferation and migration.	[138]
HRCR	HF	Inhibition of cardiac hypertrophy acting as a sponge for miRNA-223.	[139]
KCNQ1OT1	MI	Biomarker for left ventricular dysfunction.	[108]
LIPCAR	MI	Biomarker for cardiac remodeling.	[140]
LIPCAR	CHD	Biomarker.	[141]
LIPCAR	HF	Biomarker.	[140]
lincRNA-p21	CHD	Regulation of cardiomyocytes apoptosis and proliferation.	[83,142]
LINC00968	CHD	Promotion of ECs proliferation and migration acting as a sponge for miRNA-9.	[143]
lincRNA-ROR	HF	Regulation of cardiac hypertrophy acting as a sponge for miRNA-133.	[144]
Lnc-Ang362	AH	Regulation of VSMCs proliferation through miRNA-221 and -222.	[145]
Lnc-Ang362	MI	Promotion of cardiac fibrosis.	[146]
LOC285194	HF	Biomarker.	[111]
MALAT1	MI	Regulation of cardiomyocytes apoptosis and autophagy through miRNA-558; and biomarker.	[132,147,148]
MALAT1	CHD	Biomarker.	[149]
MDRL	MI	Reduction of mitochondrial fission and apoptosis acting as a sponge for miRNA-361.	[150]
MEG3	MI	Regulation of cardiomyocytes apoptosis.	[151]
MEG3	HF	Regulation of cardiac fibrosis and diastolic dysfunction.	[152]
MHRT	MI	Regulation of cardiomyocytes apoptosis; and biomarker.	[153]
MHRT	HF	Regulation of chromatin remodelers; and biomarker.	[154,155]
MIAT	MI	Regulation of cardiac hypertrophy and fibrosis acting as a sponge for miRNA-150 and -93.	[102,156,157]
MIAT	CHD	Biomarker.	[149]
MIAT	HF	Regulation of cardiac hypertrophy acting as a sponge for miRNA-150.	[157]
Mirt1/2	MI	Regulation of cardiac remodeling.	[158]
n379519	MI	Promotion of cardiac fibrosis through miRNA-30.	[159]
NEXN-AS1	CHD	Mitigation of atherosclerosis.	[160]
NONRATT021972	MI	Promotion of cardiac function.	[161]
NR_027032	AH	Biomarker.	[162]
NR_034083	AH	Biomarker.	[162]
NR_104181	AH	Biomarker.	[162]
NRF	MI	Regulation of cardiomyocytes necrosis.	[163]
NRON	HF	Biomarker.	[155]
PCFL	MI	Promotion of cardiac fibrosis through miRNA-378.	[164]
RMRP	HF	Biomarker.	[111]
RNY5	HF	Biomarker.	[111]
SMILR	CHD	Biomarker.	[165]
SOX2-OT	HF	Biomarker.	[111]
SRA1	HF	Biomarker.	[111]
TTTY15	MI	Induction of cardiomyocyte injury by hypoxia targeting miRNA-455.	[166]
UCA1	MI	Biomarker.	[167,168]
UIHTC	MI	Promotion of mitochondrial function.	[169]
Wisper	MI	Regulation of cardiac fibroblast.	[170]
ZFAS1	MI	Induction of cardiomyocyte apoptosis; cardiac contractility reduction; and biomarker.	[117,169,171]

AH, arterial hypertension; CVDs, cardiovascular diseases; CHD, coronary heart disease; ECs, endothelial cells; HF, heart failure; lncRNA, long non-coding RNA; MI, myocardial infarction; miRNA, microRNA; VSMCs, vascular smooth muscle cells; SNPs, single-nucleotide polymorphisms.

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
