# Peer review of "Long Non-Coding RNAs in Cardiovascular Diseases: Potential Function as Biomarkers and Therapeutic Targets of Exercise Training"

_ncrna, 2021, doi:10.3390/ncrna7040065_

Round 1

Reviewer 1 Report

This is a very well written review on the role of lncRNAs in cardiovascular disease, and also offers an interesting perspective into the emerging association of lncRNAs in exercise training with cardiovascular disease. The review is logically structured and discussed to an appropriate level of depth the relevant topics, from a general introduction to lncRNAs to their specific roles in different types of cardiovascular pathologies. The reference list is extensive, well balanced, and draws on recent and impactful studies in the field. Overall an excellent review that would be of great interest for readers working in this field. Some comments to further improve the review are suggested below:

- It would be useful to conclude the review with an additional section on “challenges and future perspectives”, to identify the current challenges facing the use of lncRNAs as clinical biomarkers or therapeutic targets of various cardiovascular pathologies, and how these may be addressed by future technological advances or research in progress. Are there any clinical trials commenced or about to commence in this respect? It would be relevant to comment on this and point to what to watch out for as the next major advance(s) in this field, from a translation perspective.

- For the challenges part, it would be nice to expand on this in Section 14 where the limitations and challenges are referred to but not explained in detail. Please either expand in this section or incorporate the text into the additional discussion section as suggested in the previous comment.

- The review is currently a bit too text-heavy. It would be nice to see some representative figures of experimental findings from Sections 9-13 on the roles of lncRNAs in specific cardiovascular pathologies, reproduced from a small selection of studies in each section that were published in high-quality journals and/or cited highly.

Minor comments:

- Ref 1 by WHO appears incorrectly, please revise and also include the specific identifying details of this publication.

- There are some language errors in the text, e.g. “as well as overweight and obesity” on page 1. Please check through all of the text and correct any errors.

Author Response

Combined comments from the editors and reviewers:

  • Reviewer #1 (Comments to the Author (Required)):

This is a very well written review on the role of lncRNAs in cardiovascular disease and offers an interesting perspective into the emerging association of lncRNAs in exercise training with cardiovascular disease. The review is logically structured and discussed to an appropriate level of depth the relevant topics, from a general introduction to lncRNAs to their specific roles in different types of cardiovascular pathologies. The reference list is extensive, well balanced, and draws on recent and impactful studies in the field. Overall, an excellent review that would be of great interest for readers working in this field.

  • Dear editors and reviewers,

Thank you very much for carefully reviewing our study. We appreciate your time and effort in helping us to improve our manuscript. We transcribed below all points you raised and provided our response to each of them. When appropriate, we also indicated the modifications made to our manuscript and where it will be included in the new version of the manuscript.

Minor comments:

  1. It would be useful to conclude the review with an additional section on “challenges and future perspectives”, to identify the current challenges facing the use of lncRNAs as clinical biomarkers or therapeutic targets of various cardiovascular pathologies, and how these may be addressed by future technological advances or research in progress. Are there any clinical trials commenced or about to commence in this respect? It would be relevant to comment on this and point to what to watch out for as the next major advance(s) in this field, from a translation perspective.

Response: We are grateful for this consideration. As suggested by the reviewer, we included an additional section on “lncRNAs in cardiovascular diseases: challenges and future perspectives”, to identify the current challenges facing the use of lncRNAs as clinical biomarkers or therapeutic targets of various cardiovascular pathologies, clinical trials, and how these may be addressed by future technological advances or research in progress.

  1. For the challenges part, it would be nice to expand on this in Section 14 where the limitations and challenges are referred to but not explained in detail. Please either expand in this section or incorporate the text into the additional discussion section as suggested in the previous comment.

Response: We are grateful for this consideration. As suggested by the reviewer, we incorporated the text into the additional discussion section as suggested in the previous comment.

  1. The review is currently a bit too text-heavy. It would be nice to see some representative figures of experimental findings from Sections 9-13 on the roles of lncRNAs in specific cardiovascular pathologies, reproduced from a small selection of studies in each section that were published in high-quality journals and/or cited highly.

Response: We included a representative figure of experimental findings from sections 9-13 on the roles of lncRNAs in specific cardiovascular pathologies. It was included as figure 3 in the article.

  1. Ref 1 by WHO appears incorrectly, please revise and include the specific identifying details of this publication.

Response: Thank you. The required corrections were made on WHO reference.

  1. There are some language errors in the text, e.g. “as well as overweight and obesity” on page 1. Please check through all the text and correct any errors.

Response: Thank you. The required corrections were made throughout the paper. We would like to emphasize that the paper has been submitted to MDPI for editing and proofreading (the certificate described below).

Reviewer 2 Report

The authors for sure qualify for this paper, which is a review. The manuscript is nicely written and provides great details regarding the role(s) of long non-coding RNAs in Cardiovascular diseases. However, I have some minor suggestions that can be incorporated to strengthen the review as follows:

  1. Please indicate who is the Corresponding Αuthor
  2. Abstract and elsewhere within the text. I think you should replace cardiovascular disease (CVD) with cardiovascular diseases (CVDs)
  3. Abstract: you should say “one of the leading causes of death worldwide”e
  4. P3, second paragraph (Each year….the completed knowledge”. Please provide relevant references
  5. P3, Section 4. “Biogenesis”. Please rephrase the sentence “Most of both RNAs…”
  6. Table 1. Correct the citations according to the instructions to authors
  7. Figure 3. Explain what do the arrows show.
  8. Heading 13 (and 14). Use the full name for CVD. I think that you should say “LncRNAs ad biomarkers for cardiovascular diseases”
  9. The authors should provide a few suggestions in the conclusion section

Author Response

  • Reviewer #2 (Comments to the Author (Required)):

The authors for sure qualify for this paper, which is a review. The manuscript is nicely written and provides great details regarding the role(s) of long non-coding RNAs in cardiovascular diseases. However, I have some minor suggestions that can be incorporated to strengthen the review as follows.

  1. Please indicate who is the Corresponding Αuthor.

Response: Thank you. The corresponding author has been added to the article.

  1. Abstract and elsewhere within the text. I think you should replace cardiovascular disease (CVD) with cardiovascular diseases (CVDs).

Response: Thank you. The required corrections were made throughout the paper.

  1. Abstract: you should say “one of the leading causes of death worldwide”

Response: Thank you. The required corrections were made in the abstract.

  1. P3, second paragraph (Each year….the completed knowledge”. Please provide relevant references

Response: Thank you. We included the references as suggested by the reviewer.

  1. P3, Section 4. “Biogenesis”. Please rephrase the sentence “Most of both RNAs…”

Response: Thank you. The sentence was corrected as suggested by the reviewer.

  1. Table 1. Correct the citations according to the instructions to authors

Response: Thank you. The citations were corrected in table 1.

  1. Figure 3. Explain what do the arrows show.

Response: We are grateful for this consideration. The description of the arrows was included in the figures.

  1. Heading 13 (and 14). Use the full name for CVD. I think that you should say “LncRNAs ad biomarkers for cardiovascular diseases”

Response: We are grateful for this consideration. The required corrections were made in the heading 13 and 14.

9.The authors should provide a few suggestions in the conclusion section

     Response: We are grateful for this consideration. We incorporated a few suggestions in the text into the additional discussion section.
